# Invariance and Stability
# of Deep Convolutional Representations

**Alberto Bietti**
Inria*
alberto.bietti@inria.fr

**Julien Mairal**
Inria*
julien.mairal@inria.fr

## Abstract

In this paper, we study deep signal representations that are near-invariant to groups of transformations and stable to the action of diffeomorphisms without losing signal information. This is achieved by generalizing the multilayer kernel introduced in the context of convolutional kernel networks and by studying the geometry of the corresponding reproducing kernel Hilbert space. We show that the signal representation is stable, and that models from this functional space, such as a large class of convolutional neural networks, may enjoy the same stability.

## 1   Introduction

The results achieved by deep neural networks for prediction tasks have been impressive in domains where data is structured and available in large amounts. In particular, convolutional neural networks (CNNs) [14] have shown to model well the local appearance of natural images at multiple scales, while also representing images with some invariance through pooling operations. Yet, the exact nature of this invariance and the characteristics of functional spaces where convolutional neural networks live are poorly understood; overall, these models are sometimes only seen as clever engineering black boxes that have been designed with a lot of insight collected since they were introduced.

Understanding the geometry of these functional spaces is nevertheless a fundamental question. In addition to potentially bringing new intuition about the success of deep networks, it may for instance help solving the issue of regularization, by providing ways to control the variations of prediction functions in a principled manner. Small deformations of natural signals often preserve their main characteristics, such as the class label in a classification task (*e.g.*, the same digit with different handwritings may correspond to the same images up to small deformations), and provide a much richer class of transformations than translations. Representations that are stable to small deformations allow more robust models that may exploit these invariances, which may lead to improved sample complexity. The scattering transform [5, 17] is a recent attempt to characterize convolutional multilayer architectures based on wavelets. The theory provides an elegant characterization of invariance and stability properties of signals represented via the scattering operator, through a notion of Lipschitz stability to the action of diffeomorphisms. Nevertheless, these networks do not involve "learning" in the classical sense since the filters of the networks are pre-defined, and the resulting architecture differs significantly from the most used ones.

In this work, we study these theoretical properties for more standard convolutional architectures from the point of view of positive definite kernels [27]. Specifically, we consider a functional space derived from a kernel for multi-dimensional signals, which admits a multilayer and convolutional structure that generalizes the construction of convolutional kernel networks (CKNs) [15, 16]. We show that this functional space contains a large class of CNNs with smooth homogeneous activation functions in addition to CKNs [15], allowing us to obtain theoretical results for both classes of models.

The main motivation for introducing a kernel framework is to study separately data representation and predictive models. On the one hand, we study the translation-invariance properties of the kernel representation and its stability to the action of diffeomorphisms, obtaining similar guarantees as the scattering transform [17], while preserving signal information. When the kernel is appropriately designed, we also show how to obtain signal representations that are near-invariant to the action of any group of transformations. On the other hand, we show that these stability results can be translated to predictive models by controlling their norm in the functional space. In particular, the RKHS norm controls both stability and generalization, so that stability may lead to improved sample complexity.

**Related work.** Our work relies on image representations introduced in the context of convolutional kernel networks [15, 16], which yield a sequence of spatial maps similar to traditional CNNs, but each point on the maps is possibly infinite-dimensional and lives in a reproducing kernel Hilbert space (RKHS). The extension to signals with $d$ spatial dimensions is straightforward. Since computing the corresponding Gram matrix as in classical kernel machines is computationally impractical, CKNs provide an approximation scheme consisting of learning finite-dimensional subspaces of each RKHS's layer, where the data is projected, see [15]. The resulting architecture of CKNs resembles traditional CNNs with a subspace learning interpretation and different unsupervised learning principles.

Another major source of inspiration is the study of group-invariance and stability to the action of diffeomorphisms of scattering networks [17], which introduced the main formalism and several proof techniques from harmonic analysis that were keys to our results. Our main effort was to extend them to more general CNN architectures and to the kernel framework. Invariance to groups of transformations was also studied for more classical convolutional neural networks from methodological and empirical points of view [6, 9], and for shallow learned representations [1] or kernel methods [13, 19, 22].

Note also that other techniques combining deep neural networks and kernels have been introduced. Early multilayer kernel machines appear for instance in [7, 26]. Shallow kernels for images modelling local regions were also proposed in [25], and a multilayer construction was proposed in [4]. More recently, different models based on kernels are introduced in [2, 10, 18] to gain some theoretical insight about classical multilayer neural networks, while kernels are used to define convex models for two-layer neural networks in [36]. Finally, we note that Lipschitz stability of deep models to additive perturbations was found to be important to get robustness to adversarial examples [8]. Our results show that convolutional kernel networks already enjoy such a property.

**Notation and basic mathematical tools.** A positive definite kernel $K$ that operates on a set $\mathcal{X}$ implicitly defines a reproducing kernel Hilbert space $\mathcal{H}$ of functions from $\mathcal{X}$ to $\mathbb{R}$, along with a mapping $\varphi : \mathcal{X} \to \mathcal{H}$. A *predictive model* associates to every point $z$ in $\mathcal{X}$ a label in $\mathbb{R}$; it consists of a linear function $f$ in $\mathcal{H}$ such that $f(z) = \langle f, \varphi(z) \rangle_{\mathcal{H}}$, where $\varphi(z)$ is the *data representation*. Given now two points $z, z'$ in $\mathcal{X}$, Cauchy-Schwarz's inequality allows us to control the variation of the predictive model $f$ according to the geometry induced by the Hilbert norm $\|.\|_{\mathcal{H}}$:

$$|f(z) - f(z')| \leq \|f\|_{\mathcal{H}} \|\varphi(z) - \varphi(z')\|_{\mathcal{H}}. \tag{1}$$

This property implies that two points $z$ and $z'$ that are close to each other according to the RKHS norm should lead to similar predictions, when the model $f$ has reasonably small norm in $\mathcal{H}$.

Then, we consider notation from signal processing similar to [17]. We call a signal $x$ a function in $L^2(\Omega, \mathcal{H})$, where $\Omega$ is a subset of $\mathbb{R}^d$ representing spatial coordinates, and $\mathcal{H}$ is a Hilbert space, when $\|x\|_{L^2}^2 := \int_{\Omega} \|x(u)\|_{\mathcal{H}}^2 du < \infty$, where $du$ is the Lebesgue measure on $\mathbb{R}^d$. Given a linear operator $T : L^2(\Omega, \mathcal{H}) \to L^2(\Omega, \mathcal{H}')$, the operator norm is defined as $\|T\|_{L^2(\Omega, \mathcal{H}) \to L^2(\Omega, \mathcal{H}')} := \sup_{\|x\|_{L^2(\Omega, \mathcal{H})} \leq 1} \|Tx\|_{L^2(\Omega, \mathcal{H}')}$. For the sake of clarity, we drop norm subscripts, from now on, using the notation $\| \cdot \|$ for Hilbert space norms, $L^2$ norms, and $L^2 \to L^2$ operator norms, while $| \cdot |$ denotes the Euclidean norm on $\mathbb{R}^d$. Some useful mathematical tools are also presented in Appendix A.

## 2 Construction of the Multilayer Convolutional Kernel

We now present the multilayer convolutional kernel, which operates on signals with $d$ spatial dimensions. The construction follows closely that of convolutional kernel networks [15] but generalizes it to input signals defined on the continuous domain $\Omega = \mathbb{R}^d$ (which does not prevent signals to have compact support), as done by Mallat [17] for analyzing the properties of the scattering transform; the issue of discretization where $\Omega$ is a discrete grid is addressed in Section 2.1.

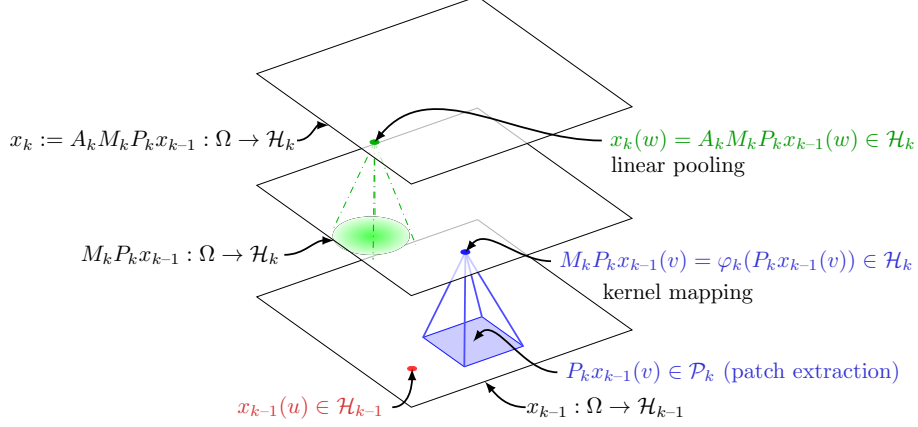

$x_k := A_k M_k P_k x_{k-1} : \Omega \to \mathcal{H}_k$

$x_k(w) = A_k M_k P_k x_{k-1}(w) \in \mathcal{H}_k$
linear pooling

$M_k P_k x_{k-1} : \Omega \to \mathcal{H}_k$

$M_k P_k x_{k-1}(v) = \varphi_k(P_k x_{k-1}(v)) \in \mathcal{H}_k$
kernel mapping

$P_k x_{k-1}(v) \in \mathcal{P}_k$ (patch extraction)

$x_{k-1}(u) \in \mathcal{H}_{k-1}$

$x_{k-1} : \Omega \to \mathcal{H}_{k-1}$

Figure 1: Construction of the $k$-th signal representation from the $k{-}1$-th one. Note that while $\Omega$ is depicted as a box in $\mathbb{R}^2$ here, our construction is supported on $\Omega = \mathbb{R}^d$. Similarly, a patch is represented as a squared box for simplicity, but it may potentially have any shape.

In what follows, an input signal is denoted by $x_0$ and lives in $L^2(\Omega, \mathcal{H}_0)$, where $\mathcal{H}_0$ is typically $\mathbb{R}^{p_0}$ (*e.g.*, with $p_0 = 3$, $x_0(u)$ may represent the RGB pixel value at location $u$). Then, we build a sequence of RKHSs $\mathcal{H}_1, \mathcal{H}_2, \ldots$, and transform $x_0$ into a sequence of "feature maps" supported on $\Omega$, respectively denoted by $x_1$ in $L^2(\Omega, \mathcal{H}_1)$, $x_2$ in $L^2(\Omega, \mathcal{H}_2)$, $\ldots$. As depicted in Figure 1, a new map $x_k$ is built from the previous one $x_{k-1}$ by applying successively three operators that perform patch extraction $(P_k)$, kernel mapping $(M_k)$ in a new RKHS $\mathcal{H}_k$, and linear pooling $(A_k)$, respectively. When going up in the hierarchy, the points $x_k(u)$ carry information from larger signal neighborhoods centered at $u$ in $\Omega$ with more invariance, as we will formally show.

**Patch extraction operator.** Given the layer $x_{k-1}$, we consider a patch shape $S_k$, defined as a compact centered subset of $\mathbb{R}^d$, *e.g.*, a box $[-1, 1] \times [-1, 1]$ for images, and we define the Hilbert space $\mathcal{P}_k := L^2(S_k, \mathcal{H}_{k-1})$ equipped with the norm $\|z\|^2 = \int_{S_k} \|z(u)\|^2 d\nu_k(u)$, where $d\nu_k$ is the normalized uniform measure on $S_k$ for every $z$ in $\mathcal{P}_k$. More precisely, we now define the linear patch extraction operator $P_k : L^2(\Omega, \mathcal{H}_{k-1}) \to L^2(\Omega, \mathcal{P}_k)$ such that for all $u$ in $\Omega$,

$$P_k x_{k-1}(u) = (v \mapsto x_{k-1}(u + v))_{v \in S_k} \in \mathcal{P}_k.$$

Note that by equipping $\mathcal{P}_k$ with a normalized measure, the operator $P_k$ preserves the norm. By Fubini's theorem, we have indeed $\|P_k x_{k-1}\| = \|x_{k-1}\|$ and hence $P_k x_{k-1}$ is in $L^2(\Omega, \mathcal{P}_k)$.

**Kernel mapping operator.** In a second stage, we map each patch of $x_{k-1}$ to a RKHS $\mathcal{H}_k$ with a kernel mapping $\varphi_k : \mathcal{P}_k \to \mathcal{H}_k$ associated to a positive definite kernel $K_k$. It is then possible to define the non-linear pointwise operator $M_k$ such that

$$M_k P_k x_{k-1}(u) := \varphi_k(P_k x_{k-1}(u)) \in \mathcal{H}_k.$$

As in [15], we use homogeneous dot-product kernels of the form

$$K_k(z, z') = \|z\|\|z'\|\kappa_k\left(\frac{\langle z, z'\rangle}{\|z\|\|z'\|}\right) \quad \text{with} \ \kappa_k(1) = 1, \tag{2}$$

which ensures that $\|M_k P_k x_{k-1}(u)\| = \|P_k x_{k-1}(u)\|$ and that $M_k P_k x_{k-1}$ is in $L^2(\Omega, \mathcal{H}_k)$. Concrete examples of kernels satisfying (2) with some other properties are presented in Appendix B.

**Pooling operator.** The last step to build the layer $x_k$ is to pool neighboring values to achieve some local shift-invariance. As in [15], we apply a linear convolution operator $A_k$ with a Gaussian kernel at scale $\sigma_k$, $h_{\sigma_k}(u) := \sigma_k^{-d} h(u/\sigma_k)$, where $h(u) = (2\pi)^{-d/2} \exp(-|u|^2/2)$. Then,

$$x_k(u) = A_k M_k P_k x_{k-1}(u) = \int_{\mathbb{R}^d} h_{\sigma_k}(u - v) M_k P_k x_{k-1}(v) dv \in \mathcal{H}_k.$$

Applying Schur's test to the integral operator $A_k$ (see Appendix A), we obtain that $\|A_k\| \le 1$. Thus, $\|x_k\| \le \|M_k P_k x_{k-1}\|$ and $x_k \in L^2(\Omega, \mathcal{H}_k)$. Note that a similar pooling operator is used in the scattering representation [5, 17], though in a different way which does not affect subsequent layers.

**Multilayer construction.** Finally, we obtain a multilayer representation by composing multiple times the previous operators. In order to increase invariance with each layer, the size of the patch $S_k$ and pooling scale $\sigma_k$ typically grow exponentially with $k$, with $\sigma_k$ and $\sup_{c \in S_k} |c|$ of the same order. With $n$ layers, the final representation is given by the feature map

$$\Phi_n(x_0) := x_n = A_n M_n P_n A_{n-1} M_{n-1} P_{n-1} \cdots A_1 M_1 P_1 x_0 \in L^2(\Omega, \mathcal{H}_n). \tag{3}$$

Then, we can define a kernel $\mathcal{K}_n$ on two signals $x_0$ and $x_0'$ by $\mathcal{K}_n(x_0, x_0') := \langle \Phi_n(x_0), \Phi_n(x_0') \rangle$, whose RKHS $\mathcal{H}_{\mathcal{K}_n}$ contains all functions of the form $f(x_0) = \langle w, \Phi_n(x_0) \rangle$ with $w \in L^2(\Omega, \mathcal{H}_n)$.

The following lemma shows that this representation preserves all information about the signal at each layer, and each feature map $x_k$ can be sampled on a discrete set with no loss of information. This suggests a natural approach for discretization which we discuss next. For space limitation reasons, all proofs in this paper are relegated to Appendix C.

**Lemma 1** (Signal preservation). *Assume that $\mathcal{H}_k$ contains linear functions $\langle w, \cdot \rangle$ with $w$ in $\mathcal{P}_k$ (this is true for all kernels $K_k$ described in Appendix B), then the signal $x_{k-1}$ can be recovered from a sampling of $x_k = A_k M_k P_k x_{k-1}$ at discrete locations as soon as the union of patches centered at these points covers all of $\Omega$. It follows that $x_k$ can be reconstructed from such a sampling.*

## 2.1 From Theory to Practice: Discretization and Signal Preservation

The previous construction defines a kernel representation for general signals in $L^2(\Omega, \mathcal{H}_0)$, which is an abstract object defined for theoretical purposes, as often done in signal processing [17]. In practice, signals are discrete, and it is thus important to discuss the problem of discretization, as done in [15]. For clarity, we limit the presentation to 1-dimensional signals ($\Omega = \mathbb{R}^d$ with $d = 1$), but the arguments can easily be extended to higher dimensions $d$ when using box-shaped patches. Notation from the previous section is preserved, but we add a bar on top of all discrete analogues of their discrete counterparts, *e.g.*, $\bar{x}_k$ is a discrete feature map in $\ell^2(\mathbb{Z}, \bar{\mathcal{H}}_k)$ for some RKHS $\bar{\mathcal{H}}_k$.

**Input signals** $x_0$ **and** $\bar{x}_0$. Discrete signals acquired by a physical device are often seen as local integrators of signals defined on a continuous domain (*e.g.*, sensors from digital cameras integrate the pointwise distribution of photons that hit a sensor in a spatial window). Let us then consider a signal $x_0$ in $L^2(\Omega, \mathcal{H}_0)$ and $s_0$ a sampling interval. By defining $\bar{x}_0$ in $\ell_2(\mathbb{Z}, \mathcal{H}_0)$ such that $\bar{x}_0[n] = x_0(ns_0)$ for all $n$ in $\mathbb{Z}$, it is thus natural to assume that $x_0 = A_0 x$, where $A_0$ is a pooling operator (local integrator) applied to an original signal $x$. The role of $A_0$ is to prevent aliasing and reduce high frequencies; typically, the scale $\sigma_0$ of $A_0$ should be of the same magnitude as $s_0$, which we choose to be $s_0 = 1$ in the following, without loss of generality. This natural assumption will be kept later in the analysis.

**Multilayer construction.** We now want to build discrete feature maps $\bar{x}_k$ in $\ell^2(\mathbb{Z}, \bar{\mathcal{H}}_k)$ at each layer $k$ involving subsampling with a factor $s_k$ w.r.t. $\bar{x}_{k-1}$. We now define the discrete analogues of the operators $P_k$ (patch extraction), $M_k$ (kernel mapping), and $A_k$ (pooling) as follows: for $n \in \mathbb{Z}$,

$$\bar{P}_k \bar{x}_{k-1}[n] := e_k^{-1/2}(\bar{x}_{k-1}[n], \bar{x}_{k-1}[n+1], \ldots, \bar{x}_{k-1}[n+e_k-1]) \in \bar{\mathcal{P}}_k := \bar{\mathcal{H}}_{k-1}^{e_k}$$

$$\bar{M}_k \bar{P}_k \bar{x}_{k-1}[n] := \bar{\varphi}_k(\bar{P}_k \bar{x}_{k-1}[n]) \in \bar{\mathcal{H}}_k$$

$$\bar{x}_k[n] = \bar{A}_k \bar{M}_k \bar{P}_k \bar{x}_{k-1}[n] := s_k^{1/2} \sum_{m \in \mathbb{Z}} \bar{h}_k[ns_k - m] \bar{M}_k \bar{P}_k \bar{x}_{k-1}[m] = (\bar{h}_k * \bar{M}_k \bar{P}_k \bar{x}_{k-1})[ns_k] \in \bar{\mathcal{H}}_k,$$

where (i) $\bar{P}_k$ extracts a patch of size $e_k$ starting at position $n$ in $\bar{x}_{k-1}[n]$ (defining a patch centered at $n$ is also possible), which lives in the Hilbert space $\bar{\mathcal{P}}_k$ defined as the direct sum of $e_k$ times $\bar{\mathcal{H}}_{k-1}$; (ii) $\bar{M}_k$ is a kernel mapping identical to the continuous case, which preserves the norm, like $M_k$; (iii) $\bar{A}_k$ performs a convolution with a Gaussian filter and a subsampling operation with factor $s_k$. The next lemma shows that under mild assumptions, this construction preserves signal information.

**Lemma 2** (Signal recovery with subsampling). *Assume that $\bar{\mathcal{H}}_k$ contains the linear functions $\langle w, \cdot \rangle$ for all $w \in \bar{\mathcal{P}}_k$ and that $e_k \geq s_k$. Then, $\bar{x}_{k-1}$ can be recovered from $\bar{x}_k$.*

We note that this result relies on recovery by deconvolution of a pooling convolution with filter $\bar{h}_k$, which is stable when its scale parameter, typically of order $s_k$ to prevent anti-aliasing, is small enough. This suggests using small values for $e_k, s_k$, as in typical recent convolutional architectures [30].

**Links between the parameters of the discrete and continuous models.** Due to subsampling, the patch size in the continuous and discrete models are related by a multiplicative factor. Specifically, a patch of size $e_k$ with discretization corresponds to a patch $S_k$ of diameter $e_k s_{k-1} s_{k-2} \ldots s_1$ in the continuous case. The same holds true for the scale parameter of the Gaussian pooling.

## 2.2 From Theory to Practice: Kernel Approximation and Convolutional Kernel Networks

Besides discretization, two modifications are required to use the image representation we have described in practice. The first one consists of using feature maps with finite spatial support, which introduces border effects that we did not study, but which are negligible when dealing with large realistic images. The second one requires finite-dimensional approximation of the kernel maps, leading to the convolutional kernel network model of [15]. Typically, each RKHS's mapping is approximated by performing a projection onto a subspace of finite dimension, a classical approach to make kernel methods work at large scale [12, 31, 34]. One advantage is its compatibility with the RKHSs (meaning that the approximations live in the respective RKHSs), and the stability results we will present next are preserved thanks to the non-expansiveness of the projection.

It is then be possible to derive theoretical results for the CKN model, which appears as a natural implementation of the kernel constructed previously; yet, we will also show in Section 5 that the results apply more broadly to CNNs that are contained in the functional space associated to the kernel.

## 3 Stability to Deformations and Translation Invariance

In this section, we study the translation-invariance and the stability of the kernel representation described in Section 2 for continuous signals under the action of diffeomorphisms. We use a similar characterization of stability to the one introduced by Mallat [17]: for a $C^1$-diffeomorphism $\tau : \Omega \to \Omega$, let $L_\tau$ denote the linear operator defined by $L_\tau x(u) = x(u - \tau(u))$, the representation $\Phi(\cdot)$ is *stable* under the action of diffeomorphisms if there exist two constants $C_1$ and $C_2$ such that

$$\|\Phi(L_\tau x) - \Phi(x)\| \le (C_1 \|\nabla \tau\|_\infty + C_2 \|\tau\|_\infty) \|x\|, \tag{4}$$

where $\nabla \tau$ is the Jacobian of $\tau$, $\|\nabla \tau\|_\infty := \sup_{u \in \Omega} \|\nabla \tau(u)\|$, and $\|\tau\|_\infty := \sup_{u \in \Omega} |\tau(u)|$. As in [17], our results will assume the regularity condition $\|\nabla \tau\|_\infty < 1/2$. In order to have a translation-invariant representation, we want $C_2$ to be small (a translation is a diffeomorphism with $\nabla \tau = 0$), and indeed we will show that $C_2$ is proportional to $1/\sigma_n$, where $\sigma_n$ is the scale of the last pooling layer, which typically increases exponentially with the number of layers $n$.

Note that unlike the scattering transform [17], we do not have a representation that preserves the norm, *i.e.*, such that $\|\Phi(x)\| = \|x\|$. While the patch extraction $P_k$ and kernel mapping $M_k$ operators do preserve the norm, the pooling operators $A_k$ may remove (or significantly reduce) frequencies from the signal that are larger than $1/\sigma_k$. Yet, natural signals such as natural images often have high energy in the low-frequency domain (the power spectra of natural images is often considered to have a polynomial decay in $1/f^2$, where $f$ is the signal frequency [33]). For such classes of signals, a large fraction of the signal energy will be preserved by the pooling operator. In particular, with some additional assumptions on the kernels $K_k$, it is possible to show [3]:

$$\|\Phi(x)\| \ge \|A_n \cdots A_0 x\|.$$

Additionally, when using a Gaussian kernel mapping $\varphi_{n+1}$ on top of the last feature map as a prediction layer instead of a linear layer, the final representation $\Phi_f(x) := \varphi_{n+1}(\Phi_n(A_0 x))$ preserves stability and always has unit norm (see the extended version of the paper [3] for details). This suggests that norm preservation may be a less relevant concern in our kernel setting.

### 3.1 Stability Results

In order to study the stability of the representation (3), we assume that the input signal $x_0$ may be written as $x_0 = A_0 x$, where $A_0$ is an initial pooling operator at scale $\sigma_0$, which allows us to control the high frequencies of the signal in the first layer. As discussed previously in Section 2.1, this assumption is natural and compatible with any physical acquisition device. Note that $\sigma_0$ can be taken arbitrarily small, making the operator $A_0$ arbitrarily close to the identity, so that this assumption does not limit the generality of our results. Moreover, we make the following assumptions for each layer $k$:

> (A1) **Norm preservation**: $\|\varphi_k(x)\| = \|x\|$ for all $x$ in $\mathcal{P}_k$;
> (A2) **Non-expansiveness**: $\|\varphi_k(x) - \varphi_k(x')\| \leq \|x - x'\|$ for all $x, x'$ in $\mathcal{P}_k$;
> (A3) **Patch sizes**: there exists $\kappa > 0$ such that at any layer $k$ we have
> $$\sup_{c \in S_k} |c| \leq \kappa \sigma_{k-1}.$$

Note that assumptions (A1-2) imply that the operators $M_k$ preserve the norm and are non-expansive. Appendix B exposes a large class of homogeneous kernels that satisfy assumptions (A1-2).

**General bound for stability.** The following result gives an upper bound on the quantity of interest, $\|\Phi(L_\tau x) - \Phi(x)\|$, in terms of the norm of various linear operators which control how $\tau$ affects each layer. The commutator of linear operators $A$ and $B$ is denoted $[A, B] = AB - BA$.

**Proposition 3.** *Let $\Phi(x) = \Phi_n(A_0 x)$ where $\Phi_n$ is defined in (3) for $x$ in $L^2(\Omega, \mathcal{H}_0)$. Then,*

$$\|\Phi(L_\tau x) - \Phi(x)\| \leq \left( \sum_{k=1}^{n} \|[P_k A_{k-1}, L_\tau]\| + \|[A_n, L_\tau]\| + \|L_\tau A_n - A_n\| \right) \|x\| \qquad (5)$$

In the case of a translation $L_\tau x(u) = L_c x(u) = x(u - c)$, it is easy to see that pooling and patch extraction operators commute with $L_c$ (this is also known as *covariance* or *equivariance* to translations), so that we are left with the term $\|L_c A_n - A_n\|$, which should control translation invariance. For general diffeomorphisms $\tau$, we no longer have exact covariance, but we show below that commutators are stable to $\tau$, in the sense that $\|[P_k A_{k-1}, L_\tau]\|$ is controlled by $\|\nabla \tau\|_\infty$, while $\|L_\tau A_n - A_n\|$ is controlled by $\|\tau\|_\infty$ and decays with the pooling size $\sigma_n$.

**Bound on $\|[P_k A_{k-1}, L_\tau]\|$.** We begin by noting that $P_k z$ can be identified with $(L_c z)_{c \in S_k}$ isometrically for all $z$ in $L^2(\Omega, \mathcal{H}_{k-1})$, since $\|P_k z\|^2 = \int_{S_k} \|L_c z\|^2 d\nu_k(c)$ by Fubini's theorem. Then,

$$\|P_k A_{k-1} L_\tau z - L_\tau P_k A_{k-1} z\|^2 = \int_{S_k} \|L_c A_{k-1} L_\tau z - L_\tau L_c A_{k-1} z\|^2 d\nu_k(c)$$
$$\leq \sup_{c \in S_k} \|L_c A_{k-1} L_\tau x - L_\tau L_c A_{k-1} z\|^2,$$

so that $\|[P_k A_{k-1}, L_\tau]\| \leq \sup_{c \in S_k} \|[L_c A_{k-1}, L_\tau]\|$. The following result lets us bound $\|[L_c A_{k-1}, L_\tau]\|$ when $|c| \leq \kappa \sigma_{k-1}$, which is satisfied under assumption (A3).

**Lemma 4.** *Let $A_\sigma$ be the pooling operator with kernel $h_\sigma(u) = \sigma^{-d} h(u/\sigma)$. If $\|\nabla \tau\|_\infty \leq 1/2$, there exists a constant $C_1$ such that for any $\sigma$ and $|c| \leq \kappa \sigma$, we have*

$$\|[L_c A_\sigma, L_\tau]\| \leq C_1 \|\nabla \tau\|_\infty,$$

*where $C_1$ depends only on $h$ and $\kappa$.*

A similar result is obtained in Mallat [17, Lemma E.1] for commutators of the form $[A_\sigma, L_\tau]$, but we extend it to handle integral operators $L_c A_\sigma$ with a shifted kernel. The proof (given in Appendix C.4) relies on the fact that $[L_c A_\sigma, L_\tau]$ is an integral operator in order to bound its norm via Schur's test. Note that $\kappa$ can be made larger, at the cost of an increase of the constant $C_1$ of the order $\kappa^{d+1}$.

**Bound on $\|L_\tau A_n - A_n\|$.** We bound the operator norm $\|L_\tau A_n - A_n\|$ in terms of $\|\tau\|_\infty$ using the following result due to Mallat [17, Lemma 2.11], with $\sigma = \sigma_n$:

**Lemma 5.** *If $\|\nabla \tau\|_\infty \leq 1/2$, we have*

$$\|L_\tau A_\sigma - A_\sigma\| \leq \frac{C_2}{\sigma} \|\tau\|_\infty, \qquad (6)$$

*with $C_2 = 2^d \cdot \|\nabla h\|_1$.*

Combining Proposition 3 with Lemmas 4 and 5, we immediately obtain the following result.

**Theorem 6.** *Let $\Phi(x)$ be a representation given by $\Phi(x) = \Phi_n(A_0 x)$ and assume (A1-3). If $\|\nabla \tau\|_\infty \leq 1/2$, we have*

$$\|\Phi(L_\tau x) - \Phi(x)\| \leq \left( C_1 (1 + n) \|\nabla \tau\|_\infty + \frac{C_2}{\sigma_n} \|\tau\|_\infty \right) \|x\|. \qquad (7)$$

This result matches the desired notion of stability in Eq. (4), with a translation-invariance factor that decays with $\sigma_n$. The dependence on a notion of depth (the number of layers $n$ here) also appears in [17], with a factor equal to the maximal length of scattering paths, and with the same condition $\|\nabla\tau\|_\infty \leq 1/2$. However, while the norm of the scattering representation is preserved as the length of these paths goes to infinity, the norm of $\Phi(x)$ can decrease with depth due to pooling layers, though this concern may be alleviated by using an additional non-linear prediction layer, as discussed previously (see also [3]).

## 3.2 Stability with Kernel Approximations

As in the analysis of the scattering transform of [17], we have characterized the stability and shift-invariance of the data representation for continuous signals, in order to give some intuition about the properties of the corresponding discrete representation, which we have described in Section 2.1.

Another approximation performed in the CKN model of [15] consists of adding projection steps on finite-dimensional subspaces of the RKHS's layers, as discusssed in Section 2.2. Interestingly, the stability properties we have obtained previously are compatible with these steps. We may indeed redefine the operator $M_k$ as the pointwise operation such that $M_k z(u) = \Pi_k \varphi_k(z(u))$ for any map $z$ in $L^2(\Omega, \mathcal{P}_k)$, instead of $M_k z(u) = \varphi_k(z(u))$; $\Pi_k : \mathcal{H}_k \to \mathcal{F}_k$ is here a projection operator onto a linear subspace. Then, $M_k$ does not necessarily preserve the norm anymore, but $\|M_k z\| \leq \|z\|$, with a loss of information corresponding to the quality of approximation of the kernel $K_k$ on the points $z(u)$. On the other hand, the non-expansiveness of $M_k$ is satisfied thanks to the non-expansiveness of the projection. Additionally, the CKN construction provides a finite-dimensional representation at each layer, which preserves the norm structure of the original Hilbert spaces isometrically. In summary, it is possible to show that the conclusions of Theorem 6 remain valid for this tractable CKN representation, but we lose signal information in the process. The stability of the predictions can then be controlled through the norm of the last (linear) layer, which is typically used as a regularizer [15].

# 4 Global Invariance to Group Actions

In Section 3, we have seen how the kernel representation of Section 2 creates invariance to translations by commuting with the action of translations at intermediate layers, and how the last pooling layer on the translation group governs the final level of invariance. It is often useful to encode invariances to different groups of transformations, such as rotations or reflections (see, *e.g.*, [9, 17, 22, 29]). Here, we show how this can be achieved by defining adapted patch extraction and pooling operators that commute with the action of a transformation group $G$ (this is known as group covariance or equivariance). We assume that $G$ is locally compact, so that we can define a left-invariant Haar measure $\mu$—that is, a measure on $G$ that satisfies $\mu(gS) = \mu(S)$ for any Borel set $S \subset G$ and $g$ in $G$. We assume the initial signal $x(u)$ is defined on $G$, and we define subsequent feature maps on the same domain. The action of an element $g \in G$ is denoted by $L_g$, where $L_g x(u) = x(g^{-1}u)$. Then, we are interested in defining a layer—that is, a succession of patch extraction, kernel mapping, and pooling operators—that commutes with $L_g$, in order to achieve equivariance to the group $G$.

**Patch extraction.** We define patch extraction as follows

$$Px(u) = (x(uv))_{v \in S} \quad \text{for all } u \in G,$$

where $S \subset G$ is a patch centered at the identity. $P$ commutes with $L_g$ since

$$PL_g x(u) = (L_g x(uv))_{v \in S} = (x(g^{-1}uv))_{v \in S} = Px(g^{-1}u) = L_g Px(u).$$

**Kernel mapping.** The pointwise operator $M$ is defined as in Section 2, and thus commutes with $L_g$.

**Pooling.** The pooling operator on the group $G$ is defined in a similar fashion as [22] by

$$Ax(u) = \int_G x(uv)h(v)d\mu(v) = \int_G x(v)h(u^{-1}v)d\mu(v),$$

where $h$ is a pooling filter typically localized around the identity element. It is easy to see from the first expression of $Ax(u)$ that $AL_g x(u) = L_g Ax(u)$, making the pooling operator $G$-equivariant.

In our analysis of stability in Section 3, we saw that inner pooling layers are useful to guarantee stability to local deformations, while global invariance is achieved mainly through the last pooling layer. In some cases, one only needs stability to a subgroup of $G$, while achieving global invariance to the whole group, *e.g.*, in the roto-translation group [21], one might want invariance to a global rotation but stability to local translations. Then, one can perform pooling just on the subgroup to stabilize (*e.g.*, translations) in intermediate layers, while pooling on the entire group at the last layer to achieve the global group invariance.

## 5   Link with Convolutional Neural Networks

In this section, we study the connection between the kernel representation defined in Section 2 and CNNs. Specifically, we show that the RKHS $\mathcal{H}_{\mathcal{K}_n}$ obtained from our kernel construction contains a set of CNNs on continuous domains with certain types of smooth homogeneous activations. An important consequence is that the stability results of previous sections apply to this class of CNNs.

**CNN maps construction.**   We now define a CNN function $f_\sigma$ that takes as input an image $x_0$ in $L^2(\Omega, \mathbb{R}^{p_0})$ with $p_0$ channels, and builds a sequence of feature maps, represented at layer $k$ as a function $z_k$ in $L^2(\Omega, \mathbb{R}^{p_k})$ with $p_k$ channels; it performs linear convolutions with a set of filters $(w_k^i)_{i=1,\dots,p_k}$, followed by a pointwise activation function $\sigma$ to obtain intermediate feature maps $\tilde{z}_k$, then applies a linear pooling filter and repeats the same operations at each layer. Note that here, each $w_k^i$ is in $L^2(S_k, \mathbb{R}^{p_{k-1}})$, with channels denoted by $w_k^{ij} \in L^2(S_k, \mathbb{R})$. Formally, the intermediate map $\tilde{z}_k$ in $L^2(\Omega, \mathbb{R}^{p_k})$ is obtained for $k \geq 1$ by

$$\tilde{z}_k^i(u) = n_k(u)\sigma\big(\langle w_k^i, P_k z_{k-1}(u) \rangle / n_k(u)\big), \tag{8}$$

where $\tilde{z}_k(u) = (\tilde{z}_k^1(u), \dots, \tilde{z}_k^{p_k}(u))$ in $\mathbb{R}^{p_k}$, and $P_k$ is the patch extraction operator, which operates here on finite-dimensional maps. The activation involves a pointwise non-linearity $\sigma$ along with a quantity $n_k(u)$ that is independent of the filters and that will be made explicit in the sequel. Finally, the map $z_k$ is obtained by using a pooling operator as in Section 2, with $z_k = A_k \tilde{z}_k$, and $z_0 = x_0$.

**Homogeneous activations.**   The choice of non-linearity $\sigma$ relies on Lemma B.2 of the appendix, which shows that for many choices of smooth functions $\sigma$, the RKHSs $\mathcal{H}_k$ defined in Section 2 contains the linear functions $z \mapsto \|z\|\sigma(\langle g, z \rangle / \|z\|)$ for all $g$ in $\mathcal{P}_k$. While this homogenization involving the quantities $\|z\|$ is not standard in classical CNNs, we note that (i) the most successful activation function, namely rectified linear units, is homogeneous—that is, $\mathrm{relu}(\langle g, z \rangle) = \|z\|\mathrm{relu}(\langle g, z \rangle / \|z\|)$; (ii) while relu is nonsmooth and thus not in our RKHSs, there exists a smoothed variant that satisfies the conditions of Lemma B.2 for useful kernels. As noticed in [35, 36], this is for instance the case for the inverse polynomial kernel described in Appendix B, In Figure 2, we plot and compare these different variants of relu. Then, we may now define the quantities $n_k(u) := \|P_k x_{k-1}(u)\|$ in (8), which are due to the homogenization, and which are independent of the filters $w_k^i$.

**Classification layer.**   The final CNN prediction function $f_\sigma$ is given by inner products with the feature maps of the last layer:

$$f_\sigma(x_0) = \langle w_{n+1}, z_n \rangle,$$

with parameters $w_{n+1}$ in $L^2(\Omega, \mathbb{R}^{p_n})$. The next result shows that for appropriate $\sigma$, the function $f_\sigma$ is in $\mathcal{H}_{\mathcal{K}_n}$. The construction of this function in the RKHS and the proof are given in Appendix D. We note that a similar construction for fully connected networks with constraints on weights and inputs was given in [35].

**Proposition 7** (CNNs and RKHSs). *Assume the activation $\sigma$ satisfies $C_\sigma(a) < \infty$ for all $a \geq 0$, where $C_\sigma$ is defined for a given kernel in Lemma B.2. Then the CNN function $f_\sigma$ defined above is in the RKHS $\mathcal{H}_{\mathcal{K}_n}$, with norm*

$$\|f_\sigma\|^2 \leq p_n \sum_{i=1}^{p_n} \|w_{n+1}^i\|_2^2 B_{n,i},$$

*where $B_{n,i}$ is defined recursively by $B_{1,i} = C_\sigma^2(\|w_1^i\|_2^2)$ and $B_{k,i} = C_\sigma^2\Big(p_{k-1}\sum_{j=1}^{p_{k-1}} \|w_k^{ij}\|_2^2 B_{k-1,j}\Big)$.*

The results of this section imply that our study of the geometry of the kernel representations, and in particular the stability and invariance properties of Section 3, apply to the generic CNNs defined

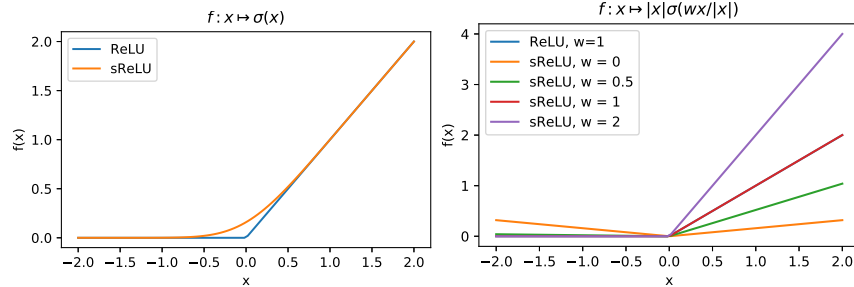

Figure 2: Comparison of one-dimensional functions obtained with relu and smoothed relu (sReLU) activations. (Left) non-homogeneous setting of [35, 36]. (Right) our homogeneous setting, for different values of the parameter $w$. Note that for $w \geq 0.5$, sReLU and ReLU are indistinguishable.

above, thanks to the Lipschitz smoothness relation (1). The smoothness is then controlled by the RKHS norm of these functions, which sheds light on the links between generalization and stability. In particular, functions with low RKHS norm (a.k.a. "large margin") are known to generalize better to unseen data (see, *e.g.*, the notion of margin bounds for SVMs [27, 28]). This implies, for instance, that generalization is harder if the task requires classifying two slightly deformed images with different labels, since this requires a function with large RKHS norm according to our stability analysis. In contrast, if a stable function (*i.e.*, with small RKHS norm) is sufficient to do well on a training set, learning becomes "easier" and few samples may be enough for good generalization.

### Acknowledgements

This work was supported by a grant from ANR (MACARON project under grant number ANR-14-CE23-0003-01), by the ERC grant number 714381 (SOLARIS project), and by the MSR-Inria joint center.

## Footnotes

*Univ. Grenoble Alpes, Inria, CNRS, Grenoble INP, LJK, 38000 Grenoble, France

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
