[Supplementary Material]

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

[2]Note that the polynomial kernel of degree $p$ can be defined here: $\kappa_{\text{poly}}(\langle z, z'\rangle) = \frac{1}{p^p}(p - 1 + \langle z, z'\rangle)^p$. For simplicity, we only consider the case $p = 2$ in this section.

[3] Note that a more precise analysis may be obtained by using finer decay bounds.

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

# A  Useful Mathematical Tools

In this section, we present preliminary mathematical tools that are used in our analysis.

**Harmonic analysis.**  We recall a classical result from harmonic analysis (see, *e.g.*, [32]), which was used many times in [17] to prove the stability of the scattering transform to the action of diffeomorphisms.

**Lemma A.1** (Schur's test). *Let $\mathcal{H}$ be a Hilbert space and $\Omega$ a subset of $\mathbb{R}^d$. Consider $T$ an integral operator with kernel $k : \Omega \times \Omega \to \mathbb{R}$, meaning that for all $u$ in $\Omega$ and $x$ in $L^2(\Omega, \mathcal{H})$,*

$$Tx(u) = \int_\Omega k(u, v)x(v)dv, \tag{9}$$

*where the integral is a Bochner integral (see, [11, 20]) when $\mathcal{H}$ is infinite-dimensional. If*

$$\forall u \in \Omega, \quad \int |k(u, v)|dv \leq C \quad and \quad \forall v \in \Omega, \quad \int |k(u, v)|du \leq C,$$

*for some constant $C$, then, $Tx$ is always in $L^2(\Omega, \mathcal{H})$ for all $x$ in $L^2(\Omega, \mathcal{H})$ and we have $\|T\| \leq C$.*

Note that while the proofs of the lemma above are typically given for real-valued functions in $L^2(\Omega, \mathbb{R})$, the result can easily be extended to Hilbert space-valued functions $x$ in $L^2(\Omega, \mathcal{H})$. In order to prove this, we consider the integral operator $|T|$ with kernel $|k|$ that operates on $L^2(\Omega, \mathbb{R}_+)$, meaning that $|T|$ is defined as in (9) by replacing $k(u, v)$ by the absolute value $|k(u, v)|$. Then, consider $x$ in $L^2(\Omega, \mathcal{H})$ and use the the triangle inequality property of Bochner integrals:

$$\|Tx\|^2 = \int_\Omega \|Tx(u)\|^2 du \leq \int_\Omega \left( \int_\Omega |k(u, v)| \|x(v)\| dv \right)^2 du = \||T||x|\|^2,$$

where the function $|x|$ is such that $|x|(u) = \|x(u)\|$ and thus $|x|$ is in $L^2(\Omega, \mathbb{R}_+)$. We may now apply Schur's test to the operator $|T|$ for real-valued functions, which gives $\||T|\| \leq C$. Then, noting that $\||x|\| = \|x\|$, we conclude with the inequality $\|Tx\|^2 \leq \||T||x|\|^2 \leq \||T|\|^2\|x\|^2 \leq C^2\|x\|^2$.

The following lemma shows that the pooling operators $A_k$ defined in Section 2 are non-expansive.

**Lemma A.2** (Non-expansiveness of pooling operators). *If $h(u) := (2\pi)^{-d/2} \exp(-|u|^2/2)$, then the pooling operator $A_\sigma$ defined for any $\sigma > 0$ by*

$$A_\sigma x(u) = \int_{\mathbb{R}^d} \sigma^{-d} h\left( \frac{u-v}{\sigma} \right) x(v)dv,$$

*has operator norm $\|A_\sigma\| \leq 1$.*

*Proof.* $A_\sigma$ is an integral operator with kernel $k(u, v) := \sigma^{-d}h((u-v)/\sigma)$. By change of variables, we have

$$\int_{\mathbb{R}^d} |k(u, v)|dv = \int_{\mathbb{R}^d} |k(u, v)|du = \int_{\mathbb{R}^d} h(u)du = 1,$$

since $h$ is a standard Gaussian and thus integrates to 1. The result follows from Schur's test. $\square$

**Kernel methods.**  We now recall a classical result that characterizes the reproducing kernel Hilbert space (RKHS) of functions defined from explicit Hilbert space mappings (see, *e.g.*, [23, §2.1]).

**Theorem A.1.** *Let $\psi : \mathcal{X} \to H$ be a feature map to a Hilbert space $H$, and let $K(z, z') := \langle \psi(z), \psi(z') \rangle_H$ for $z, z' \in \mathcal{X}$. Let $\mathcal{H}$ be the Hilbert space defined by*

$$\mathcal{H} := \{f = \langle w, \psi(\cdot) \rangle_H, w \in H\}$$
$$\|f\|_{\mathcal{H}}^2 := \inf_{w \in H} \{\|w\|_H^2, f = \langle w, \psi(\cdot) \rangle_H\}.$$

*Then $\mathcal{H}$ is the RKHS associated to kernel $K$.*

A consequence of this result is that RKHS of the kernel $\mathcal{K}_n(x, x') = \langle \Phi_n(x), \Phi_n(x') \rangle$ defined from the last layer representations $\Phi_n(x) \in L^2(\Omega, \mathcal{H}_n)$ introduced in (3) contains functions of the form $f : x \mapsto \langle w, \Phi(x) \rangle$ with $w \in L^2(\Omega, \mathcal{H}_n)$, and the RKHS norm of such a function satisfies $\|f\| \leq \|w\|_{L^2(\Omega, \mathcal{H}_n)}$.

# B   Choices of Basic Kernels

In this section, we characterize the basic kernels $K_k$ that may be used in the construction of the multilayer convolutional kernel described in Section 2. We recall here the shape of these kernels, which operate on a given Hilbert space $\mathcal{H}_0$. We consider, for $z, z'$ in $\mathcal{H}_0$,

$$K(z, z') = \|z\|\|z'\|\kappa\left(\frac{\langle z, z'\rangle}{\|z\|\|z'\|}\right), \tag{10}$$

which is positive definite when $\kappa$ admits a Maclaurin expansion with only non-negative coefficients [24, 27]—that is, $\kappa(u) = \sum_{j=0}^{+\infty} b_j u^j$ with $b_j \geq 0$ for all $j$ and all $u$ in $[-1, +1]$. Let $\varphi(\cdot)$ denote the kernel mapping associated to $K$, so that $K(z, z') = \langle \varphi(z), \varphi(z') \rangle$.

> For our stability analysis, we desire the following properties:
> - **norm preservation**: $\|\varphi(z)\| = \|z\|$; this is ensured by the condition $\kappa(1) = 1$.
> - **non-expansive mapping**: $\|\varphi(z) - \varphi(z')\| \leq \|z - z\|$.

Even though our stability results make the non-expansive assumption, they can be easily extended to Lipschitz continuous mappings. Then, the Lipschitz constants would appear in the upper-bounds from Section 3, and the stability constants would depend exponentially in the number of layers, which we would like to avoid. Below, we give a simple lemma to characterize kernels with the non-expansiveness property, and show that it applies to a large class of useful examples.

**Lemma B.1** (Non-expansive homogeneous kernel mappings). *Let $K$ be a kernel of the form (10). If $\kappa$ is convex, $\kappa(1) = 1$, and $0 \leq \kappa'(1) \leq 1$, where $\kappa'$ denotes the first derivative of $\kappa$, then the kernel mapping is non-expansive.*

*Proof.* First, we notice that

$$\|\varphi(z) - \varphi(z')\|^2 = K(z, z) + K(z', z') - 2K(z, z') = \|z\|^2 + \|z'\|^2 - 2\|z\|\|z'\|\kappa(u),$$

with $u = \langle z, z'\rangle/(\|z\|\|z'\|)$. Since $\kappa$ is convex, we also have $\kappa(u) \geq \kappa(1) + \kappa'(1)(u - 1) = 1 + \kappa'(1)(u - 1)$, and

$$
\begin{aligned}
\|\varphi(z) - \varphi(z')\|^2 &\leq \|z\|^2 + \|z'\|^2 - 2\|z\|\|z'\|\left(1 - \kappa'(1) + \kappa'(1)u\right)\\
&= (1 - \kappa'(1))\left(\|z\|^2 + \|z'\|^2 - 2\|z\|\|z'\|\right) + \kappa'(1)\left(\|z\|^2 + \|z'\|^2 - 2\langle z, z'\rangle\right)\\
&= (1 - \kappa'(1))\left|\|z\| - \|z'\|\right|^2 + \kappa'(1)\|z - z'\|^2\\
&\leq \|z - z'\|^2,
\end{aligned}
$$

where we used the fact that $0 \leq \kappa'(1) \leq 1$. Note that if we make instead the assumption that $\kappa'(1) > 1$, the same derivation shows that the kernel mapping is Lipschitz with constant $\sqrt{\kappa'(1)}$. $\square$

We are now in shape to list three a few kernels of interest that match the above assumptions. Given two vectors $z, z'$ in $\mathcal{H}_0$ with unit norm, we consider the following functions $\kappa$:

- *homogeneous Gaussian kernel*

$$\kappa_{\text{RBF}}(\langle z, z'\rangle) = e^{\alpha(\langle z, z'\rangle - 1)} = e^{-\frac{\alpha}{2}\|z - z'\|^2} \quad \text{with} \quad \alpha \leq 1.$$

  Note that if $\alpha > 1$, the kernel mapping is expansive, but is still Lipschitz with constant $\alpha$.
- *homogeneous polynomial kernel of degree $2$[2]*

$$\kappa_{\text{poly}}(\langle z, z'\rangle) = \frac{1}{4}(1 + \langle z, z'\rangle)^2.$$

- *homogeneous inverse polynomial kernel*

$$\kappa_{\text{inv-poly}}(\langle z, z'\rangle) = \frac{1}{2 - \langle z, z'\rangle}.$$

- *homogeneous arc-cosine kernel of degree 1 [7]*:

$$\kappa_{\mathrm{acos}}(\langle z, z' \rangle) = \frac{1}{\pi} \left( \sin(\theta) + (\pi - \theta) \cos(\theta) \right) \quad \text{with} \quad \theta = \arccos(\langle z, z' \rangle).$$

- *homogeneous Vovk's real polynomial kernel of degree 3*:

$$\kappa_{\mathrm{vovk}}(\langle z, z' \rangle) = \frac{1 - \langle z, z' \rangle^3}{3 - 3\langle z, z' \rangle} = \frac{1}{3} \left( 1 + \langle z, z' \rangle + \langle z, z' \rangle^2 \right).$$

For all of these kernels, it is indeed easy to see that the assumptions of Lemma B.1 are satisfied. We note that the inverse polynomial kernel was used in [35, 36]; below, we will use extend some of their results to characterize large subsets of functions that live in the corresponding RKHSs.

## B.1 Description of the RKHSs

We consider now the kernels described previously. All of them are build with a function $\kappa$ that admits a polynomial expansion $\kappa(u) = \sum_{j=0}^{+\infty} b_j u^j$ with $b_j \geq 0$ for all $j$ and all $u$ in $[-1, +1]$. We will now characterize these functional spaces by following the same strategy as [35, 36] for the non-homogeneous Gaussian and inverse polynomial kernels on Euclidean spaces. Using the previous Maclaurin expansion, we can define the following explicit feature map for any $z$ in $\mathcal{H}_0$:

$$\psi(z) = \left( \sqrt{b_0} \|z\|, \sqrt{b_1} z, \sqrt{b_2} \|z\|^{-1} z \otimes z, \sqrt{b_2} \|z\|^{-1} z \otimes z, \sqrt{b_2} \|z\|^{-1} z \otimes z \otimes z, \dots \right)$$

$$= \left( \sqrt{b_j} \|z\|^{1-j} z^{\otimes j} \right)_{j \in \mathbb{N}},$$

where $z^{\otimes j}$ denotes the tensor product of order $j$ of the vector $z$ in $\mathcal{H}_0$. Technically, the explicit mapping lives in the Hilbert space $\oplus_{j=0}^{n} \otimes^j \mathcal{H}_0$, where $\oplus$ denotes the direct sum of Hilbert spaces, and with the abuse of notation that $\otimes^0 \mathcal{H}_0$ is simply $\mathbb{R}$. Then, we have that $K(z, z') = \langle \psi(z), \psi(z') \rangle$.

The next lemma extends the results of [35, 36] to our class of kernels; it shows that the RKHS contains simple "neural network" activation functions $\sigma$ that are smooth and homogeneous.

**Lemma B.2** (Activation functions and RKHSs). *Let us consider a function $\sigma : [-1, 1] \to \mathbb{R}$ that admits a polynomial expansion $\sigma(u) := \sum_{j=0}^{\infty} a_j u^j$. Consider one of the previous kernels $K$ with explicit feature map $\psi(z) = (\sqrt{b_j} \|z\|^{1-j} z^{\otimes j})_{j \in \mathbb{N}}$, and assume that $a_j = 0$ if $b_j = 0$ for all $j$. Define the function $C_\sigma^2(\lambda^2) := \sum_{j=0}^{\infty} (a_j^2 / b_j) \lambda^{2j}$. Let $w$ such that $C_\sigma^2(\|w\|^2) < \infty$. Then, the RKHS of $K$ contains the function $f : z \mapsto \|z\| \sigma(\langle w, z \rangle / \|z\|)$, with RKHS norm $\|f\| \leq C_\sigma(\|w\|^2)$.*

*Proof.* By first considering the restriction of $K$ to unit-norm vectors $z$, the proof is very similar to [35, 36]:

$$\sigma(\langle w, z \rangle) = \sum_{j=0}^{+\infty} a_j \langle w, z \rangle^j = \sum_{j=0}^{+\infty} a_j \langle w^{\otimes j}, z^{\otimes j} \rangle = \langle \bar{w}, \psi(z) \rangle,$$

where

$$\bar{w} = \left( \frac{a_j}{\sqrt{b_j}} w^{\otimes j} \right)_{j \in \mathbb{N}}.$$

Then, the norm of $\bar{w}$ is

$$\|\bar{w}\|^2 = \sum_{j=0}^{+\infty} \frac{a_j^2}{b_j} \|w^{\otimes j}\|^2 = \sum_{j=0}^{+\infty} \frac{a_j^2}{b_j} \|w\|^{2j} = C_\sigma^2(\|w\|^2) < +\infty.$$

Using Theorem A.1, we conclude that $f$ is in the RKHS of $K$, with norm $\|f\| \leq C_\sigma(\|w\|^2)$. Finally, we extend the result to non unit-norm vectors $z$ with similar calculations and we obtain the desired result. $\qquad \square$

Some interesting activation functions are discussed in the main part of the paper, including smoothed versions of rectified linear units. The next corollary was also found to be useful in our analysis.

**Corollary B.1** (Linear functions and RKHSs). *All RKHSs considered in this section contain the linear functions of the form $z \mapsto \langle w, z \rangle$ for all $w$ in $\mathcal{H}_0$.*

# C   Proofs of Stability Results

## C.1   Proof of Lemma 1

*Proof.* We denote by $\bar{\Omega}$ the discrete set of sampling points considered in this lemma. The assumption on $\bar{\Omega}$ can be written as $\{u + v \ ; \ u \in \bar{\Omega}, v \in S_k\} = \Omega$.

Let $B$ denote an orthonormal basis of the Hilbert space $\mathcal{P}_k = L^2(S_k, \mathcal{H}_{k-1})$, and define the linear function $f_w = \langle w, \cdot \rangle \in \mathcal{H}_k$ for $w \in \mathcal{P}_k$. We thus have

$$
\begin{aligned}
P_k x_{k-1}(u) &= \sum_{w \in B} \langle w, P_k x_{k-1}(u) \rangle w \\
&= \sum_{w \in B} f_w(P_k x_{k-1}(u)) w \\
&= \sum_{w \in B} \langle f_w, M_k P_k x_{k-1}(u) \rangle w,
\end{aligned}
$$

using the reproducing property in the RKHS $\mathcal{H}_k$. Applying the pooling operator $A_k$ yields

$$
\begin{aligned}
A_k P_k x_{k-1}(u) &= \sum_{w \in B} \langle f_w, A_k M_k P_k x_{k-1}(u) \rangle w, \\
&= \sum_{w \in B} \langle f_w, x_k(u) \rangle w.
\end{aligned}
$$

Noting that $A_k P_k x = A_k (L_v x)_{v \in S_k} = (A_k L_v x)_{v \in S_k} = (L_v A_k x)_{v \in S_k} = P_k A_k x$, with $L_v x(u) := x(u + v)$, we can evaluate at $v \in S_k$ and obtain

$$
A_k x_{k-1}(u + v) = \sum_{w \in B} \langle f_w, x_k(u) \rangle w(v).
$$

Thus, taking all sampling points $u \in \bar{\Omega}$ and all $v \in S_k$, we have a full view of the signal $A_k x_{k-1}$ on all of $\Omega$ by our assumption on the set $\bar{\Omega}$.

For $f \in \mathcal{H}_{k-1}$, the signal $\langle f, x_{k-1}(u) \rangle$ can then be recovered by deconvolution as follows:

$$
\langle f, x_{k-1}(u) \rangle = \mathcal{F}^{-1} \left( \frac{\mathcal{F}(\langle f, A_k x_{k-1}(\cdot) \rangle)}{\mathcal{F}(h_{\sigma_k})} \right)(u),
$$

where $\mathcal{F}$ denotes the Fourier transform. Note that the inverse Fourier transform is well-defined here because the signal $\langle f, A_k x_k(\cdot) \rangle$ is itself a convolution with $h_{\sigma_k}$, and $\mathcal{F}(h_{\sigma_k})$ is strictly positive as the Fourier transform of a Gaussian which is also a Gaussian.

By considering all elements $f$ in an orthonormal basis of $\mathcal{H}_{k-1}$, we can recover $x_{k-1}$. $x_k$ can then be reconstructed trivially by applying operators $P_k$, $M_k$ and $A_k$.

$\square$

## C.2   Proof of Lemma 2

*Proof.* In this proof, we drop the bar notation on all quantities for simplicity; there is indeed no ambiguity since all signals are discrete here. First, we recall that $\mathcal{H}_k$ contains all linear functions on $\mathcal{P}_k = \mathcal{H}_{k-1}^{e_k}$; thus, we may consider in particular functions $f_{j,w}(z) := e_k^{1/2} \langle w, z_j \rangle$ for $j \in$

$\{1, \ldots, e_k\}$, $w \in \mathcal{H}_{k-1}$, and $z = (z_1, z_2, \ldots, z_{e_k})$ in $\mathcal{P}_k$. Then, we may evaluate

$$
\begin{aligned}
\langle f_{j,w}, s_k^{-1/2} x_k[n] \rangle &= \sum_{m \in \mathbb{Z}} h_k[ns_k - m]\langle f_{j,w}, M_k P_k x_{k-1}[m]\rangle \\
&= \sum_{m \in \mathbb{Z}} h_k[ns_k - m]\langle f_{j,w}, \varphi_k(P_k x_{k-1}[m])\rangle \\
&= \sum_{m \in \mathbb{Z}} h_k[ns_k - m] f_{j,w}(P_k x_{k-1}[m]) \\
&= \sum_{m \in \mathbb{Z}} h_k[ns_k - m]\langle w, x_{k-1}[m+j]\rangle \\
&= \sum_{m \in \mathbb{Z}} h_k[ns_k + j - m]\langle w, x_{k-1}[m]\rangle \\
&= (h_k * \langle w, x_{k-1}\rangle)[ns_k + j],
\end{aligned}
$$

where, with an abuse of notation, $\langle w, x_{k-1} \rangle$ is the real-valued discrete signal such that $\langle w, x_{k-1} \rangle[n] = \langle w, x_{k-1}[n]\rangle$. Since integers of the form $(ns_k + j)$ cover all of $\mathbb{Z}$ according to the assumption $e_k \geq s_k$, we have a full view of the signal $(h_k * \langle w, x_{k-1}\rangle)$ on $\mathbb{Z}$. We will now follow the same reasoning as in the proof of Lemma 1 to recover $\langle w, x_{k-1}\rangle$:

$$
\langle w, x_{k-1}\rangle = \mathcal{F}^{-1}\left(\frac{\mathcal{F}(h_k * \langle w, x_{k-1}\rangle)}{\mathcal{F}(h_k)}\right),
$$

where $\mathcal{F}$ is the Fourier transform. Since the signals involved there are discrete, their Fourier transform are periodic with period $2\pi$, and we note that $\mathcal{F}(h_k)$ is strictly positive and bounded away from zero. The signal $x_{k-1}$ is then recovered exactly as in the proof of Lemma 1 by considering for $w$ the elements of an orthonormal basis of the Hilbert space $\mathcal{H}_{k-1}$. $\qquad\square$

## C.3 Proof of Propositioa 3

*Proof.* Define $(MPA)_{k:j} := M_k P_k A_{k-1} M_{k-1} P_{k-1} A_{k-2} \cdots M_j P_j A_{j-1}$. Using the fact that $\|A_k\| \leq 1$, $\|P_k\| = 1$ and $M_k$ is non-expansive, we obtain

$$
\begin{aligned}
\|\Phi(L_\tau x) - \Phi(x)\| &= \|A_n(MPA)_{n:2} M_1 P_1 A_0 L_\tau x - A_n(MPA)_{n:2} M_1 P_1 A_0 x\| \\
&\leq \|A_n(MPA)_{n:2} M_1 P_1 A_0 L_\tau x - A_n(MPA)_{n:2} M_1 L_\tau P_1 A_0 x\| \\
&\quad + \|A_n(MPA)_{n:2} M_1 L_\tau P_1 A_0 x - A_n(MPA)_{n:2} M_1 P_1 A_0 x\| \\
&\leq \|[P_1 A_0, L_\tau]\|\|x\| \\
&\quad + \|A_n(MPA)_{n:2} M_1 L_\tau P_1 A_0 x - A_n(MPA)_{n:2} M_1 P_1 A_0 x\|.
\end{aligned}
$$

Note that $M_1$ is defined point-wise, and thus commutes with $L_\tau$:

$$
M_1 L_\tau x(u) = \varphi_1(L_\tau x(u)) = \varphi_1(x(u - \tau(u)) = M_1 x(u - \tau(u)) = L_\tau M_1 x(u).
$$

By noticing that $\|M_1 P_1 A_0 x\| \leq \|x\|$, we can expand the second term above in the same way. Repeating this by induction yields

$$
\begin{aligned}
\|\Phi(L_\tau x) - \Phi(x)\| &\leq \sum_{k=1}^{n} \|[P_k A_{k-1}, L_\tau]\|\|x\| + \|A_n L_\tau (MPA)_{n:1} x - A_n(MPA)_{n:1} x\| \\
&\leq \sum_{k=1}^{n} \|[P_k A_{k-1}, L_\tau]\|\|x\| + \|A_n L_\tau - A_n\|\|x\|,
\end{aligned}
$$

and the result follows by decomposing $A_n L_\tau = [A_n, L_\tau] + L_\tau A_n$ using the triangle's inequality. $\qquad\square$

## C.4 Proof of Lemma 4

*Proof.* The proof follows in large parts the methodology introduced by Mallat [17] in the analysis of the stability of the scattering transform. More precisely, we will follow in part the proof of Lemma

E.1 of [17]. The kernel (in the sense of Lemma A.1) of $A_\sigma$ is $h_\sigma(z - u) = \sigma^{-d} h(\frac{z-u}{\sigma})$. Throughout the proof, we will use the following bounds on the decay of $h$ for simplicity, [3] as in [17]:

$$|h(u)| \leq \frac{C_h}{(1 + |u|)^{d+2}}$$

$$|\nabla h(u)| \leq \frac{C'_h}{(1 + |u|)^{d+2}},$$

which are satisfied for the Gaussian function $h$ thanks to its exponential decay.

We now decompose the commutator

$$[L_c A_\sigma, L_\tau] = L_c A_\sigma L_\tau - L_\tau L_c A_\sigma = L_c (A_\sigma - L_c^{-1} L_\tau L_c A_\sigma L_\tau^{-1}) L_\tau = L_c T L_\tau,$$

with $T := A_\sigma - L_c^{-1} L_\tau L_c A_\sigma L_\tau^{-1}$. Hence,

$$\|[L_c A_\sigma, L_\tau]\| \leq \|L_c\| \|L_\tau\| \|T\|.$$

We have $\|L_c\| = 1$ since the translation operator $L_c$ preserves the norm. Note that we have

$$2^{-d} \leq (1 - \|\nabla \tau\|_\infty)^d \leq \det(I - \nabla \tau(u)) \leq (1 + \|\nabla \tau\|_\infty)^d \leq 2^d, \tag{11}$$

for all $u \in \Omega$. Thus, for $f \in L^2(\Omega)$,

$$\|L_\tau f\|^2 = \int_\Omega |f(z - \tau(z))|^2 dz = \int_\Omega |f(u)|^2 \det(I - \nabla \tau(u))^{-1} du$$
$$\leq (1 - \|\nabla \tau\|_\infty)^{-d} \|f\|^2,$$

such that $\|L_\tau\| \leq (1 - \|\nabla \tau\|_\infty)^{-d/2} \leq 2^{d/2}$. This yields

$$\|[L_c A_\sigma, L_\tau]\| \leq 2^{d/2} \|T\|.$$

**Kernel of $T$.** We now show that $T$ is an integral operator and describe its kernel. Let $\xi = (I - \tau)^{-1}$, so that $L_\tau^{-1} f(z) = f(\xi(z))$ for any function $f$ in $L^2(\Omega)$. We have

$$A_\sigma L_\tau^{-1} f(z) = \int h_\sigma(z - v) f(\xi(v)) dv$$
$$= \int h_\sigma(z - u + \tau(u)) f(u) \det(I - \nabla \tau(u)), du$$

using the change of variable $v = u - \tau(u)$, giving $\left|\frac{dv}{du}\right| = \det(I - \nabla \tau(u))$. Then note that $L_c^{-1} L_\tau L_c f(z) = L_\tau L_c f(z + c) = L_c f(z + c - \tau(z + c)) = f(z - \tau(z + c))$. This yields the following kernel for the operator $T$:

$$k(z, u) = h_\sigma(z - u) - h_\sigma(z - \tau(z + c) - u + \tau(u)) \det(I - \nabla \tau(u)). \tag{12}$$

A similar operator appears in Lemma E.1 of [17], whose kernel is identical to (12) when $c = 0$.

As in [17], we decompose $T = T_1 + T_2$, with kernels

$$k_1(z, u) = h_\sigma(z - u) - h_\sigma((I - \nabla \tau(u))(z - u)) \det(I - \nabla \tau(u))$$
$$k_2(z, u) = \det(I - \nabla \tau(u)) \left( h_\sigma((I - \nabla \tau(u))(z - u)) - h_\sigma(z - \tau(z + c) - u + \tau(u)) \right).$$

The kernel $k_1(z, u)$ appears in [17], whereas the kernel $k_2(z, u)$ involves a shift $c$ which is not present in [17]. For completeness, we include the proof of the bound for both operators, even though only dealing with $k_2$ requires slightly new developments.

**Bound on $\|T_1\|$.** We can write $k_1(z, u) = \sigma^{-d} g(u, (z - u)/\sigma)$ with

$$g(u, v) = h(v) - h((I - \nabla \tau(u))v) \det(I - \nabla \tau(u))$$
$$= (1 - \det(I - \nabla \tau(u))) h((I - \nabla \tau(u))v) + h(v) - h((I - \nabla \tau(u))v).$$

Using the fundamental theorem of calculus on $h$, we have

$$h(v) - h((I - \nabla\tau(u))v) = \int_0^1 \langle \nabla h((I + (t-1)\nabla\tau(u))v), \nabla\tau(u)v \rangle dt.$$

Noticing that

$$|(I + (t-1)\nabla\tau(u))v| \geq (1 - \|\nabla\tau\|_\infty)|v| \geq (1/2)|v|,$$

and that $\det(I - \nabla\tau(u))) \geq (1 - \|\nabla\tau\|_\infty)^d \geq 1 - d\|\nabla\tau\|_\infty$, we bound each term as follows

$$|(1 - \det(I - \nabla\tau(u)))h((I - \nabla\tau(u))v)| \leq d\|\nabla\tau\|_\infty \frac{C_h}{(1 + \frac{1}{2}|v|)^{d+2}}$$

$$\left| \int_0^1 \langle \nabla h((I + (t-1)\nabla\tau(u))v), \nabla\tau(u)v \rangle dt \right| \leq \|\nabla\tau\|_\infty \frac{C_h'|v|}{(1 + \frac{1}{2}|v|)^{d+2}}.$$

We thus have

$$|g(u,v)| \leq \|\nabla\tau\|_\infty \frac{C_h d + C_h'|v|}{(1 + \frac{1}{2}|v|)^{d+2}}.$$

Using appropriate changes of variables in order to bound $\int |k_1(z,u)|du$ and $\int |k_1(z,u)dz|$, Schur's test yields

$$\|T_1\| \leq C_1 \|\nabla\tau\|_\infty, \tag{13}$$

with

$$C_1 = \int_\Omega \frac{C_h d + C_h'|v|}{(1 + \frac{1}{2}|v|)^{d+2}} dv$$

**Bound on $\|T_2\|$.** Let $\alpha(z,u) = \tau(z+c) - \tau(u) - \nabla\tau(u)(z-u)$, and note that we have

$$\begin{aligned}
|\alpha(z,u)| &\leq |\tau(z+c) - \tau(u)| + |\nabla\tau(u)(z-u)| \\
&\leq \|\nabla\tau\|_\infty|z + c - u| + \|\nabla\tau\|_\infty|z - u| \\
&\leq \|\nabla\tau\|_\infty(|c| + 2|z - u|).
\end{aligned} \tag{14}$$

The fundamental theorem of calculus yields

$$k_2(z,u) = -\det(I - \nabla\tau(u)) \int_0^1 \langle \nabla h_\sigma(z - \tau(z+c) - u + \tau(u) - t\alpha(z,u)), \alpha(z,u) \rangle dt.$$

We note that $|\det(I - \nabla\tau(u))| \leq 2^d$, and $\nabla h_\sigma(v) = \sigma^{-d-1}\nabla h(v/\sigma)$. Using the change of variable $z' = (z - u)/\sigma$, we obtain

$$\int |k_2(z,u)|dz \leq$$

$$2^d \int \int_0^1 \left| \nabla h\left(z' + \frac{\tau(u + \sigma z' + c) - \tau(u) - t\alpha(u + \sigma z', u)}{\sigma}\right) \right| \left| \frac{\alpha(u + \sigma z', u)}{\sigma} \right| dt dz'.$$

We can use the upper bound (14):

$$\left| \frac{\alpha(u + \sigma z', u)}{\sigma} \right| \leq \|\nabla\tau\|_\infty(\kappa + 2|z'|). \tag{15}$$

Separately, we have $|\nabla h(v(z'))| \leq C_h'/(1 + |v(z')|)^{d+2}$, with

$$v(z') := z' + \frac{\tau(u + \sigma z' + c) - \tau(u) - t\alpha(u + \sigma z', u)}{\sigma}.$$

For $|z'| > 2\kappa$, we have

$$\begin{aligned}
\left| \frac{\tau(u + \sigma z' + c) - \tau(u) - t\alpha(u + \sigma z', u)}{\sigma} \right| &= \left| t\nabla\tau(u)z' + (1-t)\frac{\tau(u + \sigma z' + c) - \tau(u)}{\sigma} \right| \\
&\leq t\|\nabla\tau\|_\infty|z'| + (1-t)\|\nabla\tau\|_\infty(|z'| + \kappa) \\
&\leq \frac{3}{2}\|\nabla\tau\|_\infty|z'| \leq \frac{3}{4}|z'|,
\end{aligned}$$

and hence, using the reverse triangle inequality, $|v(z')| \geq |z'| - \frac{3}{4}|z'| = \frac{1}{4}|z'|$. This yields the upper bound

$$|\nabla h(v(z'))| \leq \begin{cases} C_h', & \text{if } |z'| \leq 2\kappa \\ \frac{C_h'}{(1+\frac{1}{4}|z'|)^{d+2}}, & \text{if } |z'| > 2\kappa. \end{cases} \tag{16}$$

Combining these two bounds, we obtain

$$\int |k_2(z,u)|dz \leq C_2\|\nabla\tau\|_\infty,$$

with

$$C_2 := 2^d C_h' \left( \int_{|z'|<2\kappa} (\kappa + 2|z'|)dz' + \int_{|z'|>2\kappa} \frac{\kappa + 2|z'|}{(1+\frac{1}{4}|z'|)^{d+2}} dz' \right).$$

Note that the dependence of the first integral on $\kappa$ is of order $k^{d+1}$. Following the same steps with the change of variable $u' = (z-u)/\sigma$, we obtain the bound $\int |k_2(z,u)|du \leq C_2\|\nabla\tau\|_\infty$. Schur's test then yields

$$\|T_2\| \leq C_2\|\nabla\tau\|_\infty. \tag{17}$$

We have thus proven

$$\|[L_c A_\sigma, L_\tau]\| \leq 2^{d/2}\|T\| \leq 2^{d/2}(C_1 + C_2)\|\nabla\tau\|_\infty.$$

$\square$

## D  Proof of Proposition 7 and Construction of CNNs in the RKHS

In this section, we describe the space of functions (RKHS) $\mathcal{H}_{\mathcal{K}_n}$ associated to the kernel $\mathcal{K}_n(x,x') = \langle \Phi_n(x), \Phi_n(x') \rangle$ where $\Phi_n$ is the final representation of Eq. (3), in particular showing it contains the set of CNNs with activations described in Section B.1.

**Construction of a CNN in the RKHS.**  Let us consider the definition of the CNN presented in Section 5. We will show that it can be seen as a point in the RKHS of $\mathcal{K}_n$. According to Lemma B.2, we consider $\mathcal{H}_k$ that contains all functions of the form $z \in \mathcal{P}_k \mapsto \|z\|\sigma(\langle w, z \rangle/\|z\|)$, with $w \in \mathcal{P}_k$.

Then, we define the initial quantities $f_1^i \in \mathcal{H}_1, g_1^i \in \mathcal{P}_1$ for $i = 1, \ldots, p_1$ such that

$$g_1^i = w_1^i \in L^2(S_1, \mathbb{R}^{p_0}) = L^2(S_1, \mathcal{H}_0) = \mathcal{P}_1$$
$$f_1^i(z) = \|z\|\sigma(\langle g_i^0, z \rangle/\|z\|) \quad \text{for } z \in \mathcal{P}_1,$$

and we recursively define, from layer $k$–1, the quantities $f_k^i \in \mathcal{H}_k, g_k^i \in \mathcal{P}_k$ for $i = 1, \ldots, p_k$:

$$g_k^i(v) = \sum_{j=1}^{p_{k-1}} w_k^{ij}(v)f_{k-1}^j \quad \text{where} \quad w_k^i(v) = (w_k^{ij}(v))_{j=1,\ldots,p_{k-1}}$$
$$f_k^i(z) = \|z\|\sigma(\langle g_k^i, z \rangle/\|z\|) \quad \text{for } z \in \mathcal{P}_k.$$

Then, we will show that $\tilde{z}_k^i(u) = f_k^i(P_k x_{k-1}(u)) = \langle f_k^i, M_k P_k x_{k-1}(u) \rangle$, which correspond to feature maps at layer $k$ and index $i$ in a CNN. Indeed, this is easy to see for $k = 1$ by construction with

filters $w_1^i(v)$, and for $k \geq 2$, we have

$$
\begin{aligned}
\tilde{z}_k^i(u) &= n_k(u)\sigma\big(\langle w_k^i, P_k z_{k-1}(u)\rangle / n_k(u)\big) \\
&= n_k(u)\sigma\big(\langle w_k^i, P_k A_{k-1}\tilde{z}_{k-1}(u)\rangle / n_k(u)\big) \\
&= n_k(u)\sigma\left(\frac{1}{n_k(u)} \sum_{j=1}^{p_{k-1}} \int_{S_k} w_k^{ij}(v) A_{k-1}\tilde{z}_{k-1}^j(u+v) d\nu_k(v)\right) \\
&= n_k(u)\sigma\left(\frac{1}{n_k(u)} \sum_{j=1}^{p_{k-1}} \int_{S_k} w_k^{ij}(v)\langle f_{k-1}^j, A_{k-1}M_{k-1}P_{k-1}x_{k-2}(u+v)\rangle d\nu_k(v)\right) \\
&= n_k(u)\sigma\left(\frac{1}{n_k(u)} \int_{S_k} \langle g_k^i(v), A_{k-1}M_{k-1}P_{k-1}x_{k-2}(u+v)\rangle d\nu_k(v)\right) \\
&= n_k(u)\sigma\left(\frac{1}{n_k(u)} \int_{S_k} \langle g_k^i(v), x_{k-1}(u+v)\rangle d\nu_k(v)\right) \\
&= n_k(u)\sigma\left(\frac{1}{n_k(u)} \langle g_k^i(v), P_k x_{k-1}(u)\rangle\right) \\
&= f_k^i(P_k x_{k-1}(u)),
\end{aligned}
$$

where $n_k(u) := \|P_k x_{k-1}(u)\|$. Note that we have used many times the fact that $A_k$ operates on each channel independently when applied to a finite-dimensional map.

The final prediction function is of the form $f_\sigma(x_0) = \langle w_{n+1}, z_n\rangle$ with $w_{n+1}$ in $L^2(\Omega, \mathbb{R}^{p_n})$. Then, we can define the following function $g_\sigma$ in $L^2(\Omega, \mathcal{H}_n)$ such that

$$
g_\sigma(u) = \sum_{j=1}^{p_n} w_{n+1}^j(u) f_n^j,
$$

which yields

$$
\begin{aligned}
\langle g_\sigma, x_n\rangle &= \sum_{j=1}^{p_n} \int_\Omega w_{n+1}^j(u)\langle f_n^j, x_n(u)\rangle du \\
&= \sum_{j=1}^{p_n} \int_\Omega w_{n+1}^j(u)\langle f_n^j, A_n M_n P_n x_{n-1}(u)\rangle du \\
&= \sum_{j=1}^{p_n} \int_\Omega w_{n+1}^j(u) A_n \tilde{z}_n^j(u) du \\
&= \sum_{j=1}^{p_n} \int_\Omega w_{n+1}^j(u) z_n^j(u) du \\
&= \sum_{j=1}^{p_n} \langle w_{n+1}^j, z_n^j\rangle = f_\sigma(x_0),
\end{aligned}
$$

which corresponds to a linear layer after pooling. Since the RKHS of $\mathcal{K}_n$ contains all functions of the form $f(x_0) = \langle g, \Phi_n(x_0)\rangle = \langle g, x_n\rangle$, for $g$ in $L^2(\Omega, \mathcal{H}_n)$, we have that $f_\sigma$ is in the RKHS.

**Norm of the CNN.** As shown in Section B.1, the RKHS norm of a function $f : z \in \mathcal{P}_k \mapsto \|z\|\sigma(\langle w, z\rangle / \|z\|)$ in $\mathcal{H}_k$ is bounded by $C_\sigma(\|w\|^2)$, where $C_\sigma$ depends on the activation $\sigma$. We then

have

$$\|f_1^i\|^2 \leq C_\sigma^2(\|w_1^i\|_2^2) \quad \text{where} \quad \|w_1^i\|_2^2 = \int_{S_1} \|w_1^i(v)\| d\nu_1(v)$$

$$\|f_k^i\|^2 \leq C_\sigma^2(\|g_k^i\|^2)$$

$$\|g_k^i\|^2 = \int_{S_k} \|\sum_{j=1}^{p_{k-1}} w_k^{ij}(v) f_{k-1}^j\|^2 d\nu_k(v)$$

$$\leq p_{k-1} \sum_{j=1}^{p_{k-1}} \left( \int_{S_k} |w_k^{ij}(v)|^2 d\nu_k(v) \right) \|f_{k-1}^j\|^2$$

$$= p_{k-1} \sum_{j=1}^{p_{k-1}} \|w_k^{ij}\|_2^2 \|f_{k-1}^j\|^2,$$

where in the last inequality we use $\|a_1 + \ldots + a_n\|^2 \leq n(\|a_1\|^2 + \ldots + \|a_n\|^2)$. Since $C_\sigma^2$ is monotonically increasing (typically exponentially in its argument), we have for $k = 1, \ldots, n-1$ the recursive relation

$$\|f_k^i\|^2 \leq C_\sigma^2 \left( p_{k-1} \sum_{j=1}^{p_{k-1}} \|w_k^{ij}\|_2^2 \|f_{k-1}^j\|^2 \right).$$

The norm of the final prediction function $f \in L^2(\Omega, \mathcal{H}_n)$ is bounded as follows, using similar arguments as well as Theorem A.1:

$$\|f_\sigma\|^2 \leq \|w_{n+1}\|^2 \leq p_n \sum_{j=1}^{p_n} \left( \int_\Omega |w_{n+1}^j(u)|^2 du \right) \|f_n^j\|^2.$$