[Reviews · NeurIPS 2017]

Reviewer 1



The paper studied some nice properties about deep convolutional kernel networks, i.e.CKNs are near invariant to groups of transformation and stable to the action of diffeomorohisms, if appropriate patch extraction operator´╝îkernel mapping operator and pooling operator are carefully designed. Authors also showed that under certain assumption, the nice properties obtained from CKNs also applies to CNNs. In general, the paper is well written and technically sound. The only weakness is that the paper does not provide any empirical experiments to validate the theoretical analysis. I wish to see some experimental results showing the conclusion that CKNs with appropriate settings are invariant to the transformation and diffeomorphism. Another question is whether random projection or explicit kernel mapping with finite number of Fourier series can also be a candidate solution for the kernel approximation. If so, what is the effect on the stability.

Reviewer 2



The primary focus of the paper is CKN (convolutional kernel network) [13, 14]. In this manuscript the authors analyse the stability [w.r.t. C^1 diffeomorphisms (such as translation), in the sense of Eq. (4)] of the representation formed by CKNs. They show that for norm-preserving and non-expansive kernels [(A1-A2) in line 193] stability holds for appropriately chosen patch sizes [(A3)]. Extension from (R^d,+) to locally compact groups is sketched in Section 4. The paper is nicely organized, clearly written, technically sound, combining ideas from two exciting areas (deep networks and kernels). The stability result can be of interest to the ML community. -The submission would benefit from adding further motivation on the stability analysis. Currently there is only one short sentence (line 56-57: 'Finally, we note that the Lipschitz stability of deep predictive models was found to be important to get robustness to adversarial examples [7].') which motivates the main contribution of the paper. -Overloading the \kappa notation in (A3) [line 193] might be confusing, it also denotes a function defining kernel K in Eq. (10). -In the displayed equation between line 384 and 385, the second part ('and \forall v...') is superfluous; given the symmetry of kernel k, it is identical to the first constraint ('\forall u ...'). -Line 416: the definition of \phi is missing, it should be introduced in Eq. (10) [=<\phi(z),\phi(z')>_{H(K)}]. -Line 427-428: The inequality under '=' seems to also hold with equality, | ||z|| - ||z|| |^2 should be | ||z|| - ||z'|| |^2. References: [3,6,8,13,14,18,25-27,30,31]: page information is missing. [9]: appeared -> Amit Daniely, Roy Frostig, Yoram Singer. Toward Deeper Understanding of Neural Networks: The Power of Initialization and a Dual View on Expressivity. Advances in Neural Information Processing Systems (NIPS), pages 2253-2261, 2016. [17]: appeared -> Krikamol Muandet, Kenji Fukumizu, Bharath Sriperumbudur, Bernhard Sch{\"o}lkopf. Kernel Mean Embedding of Distributions: A Review and Beyond. Foundations and Trends in Machine Learning, 10(1-2): 1-141. [19]: appeared -> Anant Raj, Abhishek Kumar, Youssef Mroueh, Tom Fletcher, Bernhard Sch{\"o}lkopf. International Conference on Artificial Intelligence and Statistics (AISTATS), PMLR 54:1225-1235, 2017. [32]: accepted (https://2017.icml.cc/Conferences/2017/Schedule?type=Poster) -> Yuchen Zhang, Percy Liang, Martin Wainwright. Convexified Convolutional Neural Networks. International Conference on Machine Learning (ICML), 2017, accepted.

Reviewer 3



The paper is well written and rigorous and it successfully shows how (some type of) functional spaces implemented by DCNNs can be described in terms of multilayer kernels. It also shows how the architecture has nice properties: (almost) invariance to group transformations and stability to diffeomorphic transformations. I recommend the acceptance given the comments below: *To the reviewer's understanding the main paper contribution is a rewriting of CNNS architecture in the context of RKHS. Please add in the text some comments why this reformulation is/will be useful (e.g. kernel formulation may lead to interesting results on generalization bounds?, is it computationally more efficient to implement using standard techniques/approximation on kernels?) More in general is the novelty of the approach going beyond a reformulation of DCNNs in the context of RKHS? Also clarify the novelty w.r.t. the paper "Convolutional Kernel Networks" of Julien Mairal, Piotr Koniusz, Zaid Harchaoui, and Cordelia Schmid. * The authors stress the importance of "not loosing the signal information". Why is so? DCNNs are often used in relation to some tasks (e.g. image classification). Unless the task is signal reconstruction only task-relevant information need to be preserved. Please comment on this. * Regarding group invariance: is there any connection with the invariant kernel approach adopted by Hans Burkhardt (invariant kernels)? * More of a curiosity: you define patch, pooling and kernel maps that commute with the group action. Will any other operator (in the set of operators that commutes with all group elements) ensure the equivariance property of DCNN w.r.t. the group transformations?